# Structure and Properties of Amorphous Quasi-High-Entropy Fe-Co-Ni-Cr-(Mo,V)-B Alloys with Various Boron Content

Andrey Bazlov [1,2,*], Ilia Strochko [1,2], Evgeny Ubyivovk [1], Mark Parkhomenko [1,2], Daria Magomedova [1] and Erzhena Zanaeva [1,2]

1 Laboratory for Mechanics of Advanced Bulk Nanomaterials for Innovative Engineering Applications, St. Petersburg State University, St. Petersburg 199034, Russia

2 Department of Physical Metallurgy of Non-Ferrous Metals, NUST MISIS, 4 Leninskiy Prospekt, Moscow 119049, Russia

* Correspondence: bazlov@misis.ru

**Abstract:** This study focuses on the effect of replacing molybdenum with vanadium in rapidly quenched quasi-high-entropy alloys of the Fe-Co-Ni-Cr-(Mo,V)-B system. The paper analyzes the effect of the chemical composition of alloys with different boron content levels on structure formation, characteristic temperatures of alloys, and mechanical properties. An analysis of the relationship between the structure of alloys and their properties is performed. X-ray diffraction and scanning electron microscopy methods were used in the work to establish the structural dependencies. Characteristic temperatures were determined by differential scanning calorimetry. It is suggested that the addition of vanadium to the alloys of this system leads to the formation of vanadium nitrides in the structure, due to the binding of dissolved nitrogen. Furthermore, it is found that replacing molybdenum with vanadium leads to an increase in the thermal stability of the amorphous phase.

**Keywords:** amorphous alloys; high-entropy alloys; microhardness; microstructure; XRD; SEM; TEM





## 1. Introduction

Amorphous alloys, first obtained by Duwez in 1960 [1], present the most metastable structural state, which is similar to the structure of a liquid. The absence of a long-range order in the arrangement of atoms is the reason for the unique mechanical and physical properties of these materials [2]. The development of alloy compositions that allow us to obtain an amorphous state in ingots with large cross-sections has facilitated the wide industrial use of metallic glasses [3]. The amorphous structure of alloys is thermally unstable and, upon heating, tends to transform into a stable crystalline state [4,5]. Quite often, structural transformations of the amorphous phase passes through high-temperature metastable states [6].

The miniaturization of electronic devices has led to a demand for microelectromechanical systems (MEMSs)—devices that combine mechanical devices and electronic components in one integral package [7,8]. Due to their small size (several tens of microns), materials with high specific strength and a homogeneous structure are in demand for the mechanical part of these devices [9]. The use of conventional crystalline alloys is impossible in MEMSs since, at the microlevel, most alloys have structural inhomogeneity, such as the strengthening phases and defects of the crystalline structure, which can have a negative effect on the mechanical properties of microparts. At present, single-crystal silicon is used to manufacture such parts [10]; however, crystallographic anisotropy is a major disadvantage of single-crystal silicon. Amorphous alloys are promising materials for the manufacture of the mechanical parts of MEMSs, which not only have high strength levels, but also, due to the absence of a crystal structure and its defects, have homogeneous (isotropic) properties at the micro- and sub-microlevels [2,11].

Amorphous iron-based alloys have been well known since the 1980s and have found wide applications in the electronic and electrical industries [12–14]. This became possible due to their unique set of magnetic properties [15–17]. In addition, iron-based amorphous alloys have high-strength properties [18–21]. Industrial, amorphous, soft, magnetic iron-based alloys have a relatively low glass-forming ability; their critical thickness is about 50–100 μm [22]. Iron-based bulk metallic glasses have extremely high brittleness levels, even in the ribbon form [18]. An increase in the glass-forming ability is possible by applying a high-entropy approach [23]—replacing iron with other d-elements, such as cobalt, nickel, chromium, and others [24–27]. The high-entropy effect [28] hinders the crystallization of a multicomponent solid solution from melting due to an increase in the size of the critical nucleus [29], reduces the critical cooling rate for the formation of an amorphous structure, and increases the glass-forming ability of the alloys. In the first studies on the application of the high-entropy approach, iron was replaced by cobalt and nickel in order to preserve the ferromagnetic state of the material in the amorphous state at room temperature [23,25]. Moreover, the addition of chromium to the composition significantly increased the corrosion resistance of alloys [25,30]. It was shown that the partial replacement of Fe, Ni, and Co atoms by molybdenum atoms led to a significant increase in the strength of alloys in the (Fe,Co,Ni,Cr,Mo)B system while maintaining technological plasticity [31]. However, these alloys have an unusual feature, a decrease in strength properties with an increase in the boron content from 11 to 17% due to an increase in the interatomic distance [32].

There is lack of data on the vanadium addition effect on crystalline high-entropy alloys based on the Fe-Co-Ni-Cr system and to amorphous alloys based on the Fe-Co-Ni-Cr-B system. Simultaneously, there are data on the effect of vanadium on the structure and mechanical properties of high-entropy alloys obtained by magnetron sputtering [33]. It has been established that the addition of V to CoCrFeNiMn alloys that do not contain metalloids promotes the formation of an amorphous structure and the appearance of nanotwins during deformation, which contribute to the effect of strain hardening. Vanadium has average atomic radii between Fe and Mo, which should allow for the formation of a denser atomic structure of the amorphous phase and improve the mechanical properties of the alloys while maintaining the glass-forming ability. In this work, the effect of vanadium addition on the structure formation and mechanical properties of alloys $(Fe_{0.25}Ni_{0.25}Co_{0.25}Cr_{0.125}Mo_{0.125-x}V_x)_{100-y}B_y$, where x = 0; 0.0625; 0.125, y = 11–17, is studied. Moreover, based on the obtained results, we assume that these additions lead to the formation of refractory vanadium-rich nitrides in the structures of the studied alloy ribbons.

## 2. Materials and Methods

Alloy ingots with a nominal composition $(Fe_{0.25}Ni_{0.25}Co_{0.25}Cr_{0.125}Mo_{0.125-x}V_x)_{100-y}B_y$, where x = 0; 0.0625; 0.125 and y = 11; 12; 14; 17, were prepared by arc melting pure Fe, Ni, Co, Cr, and V metals (99.9 wt.%), pre-alloyed Ni-50Mo (wt.%), and crystalline boron, 99.8% pure. To obtain a homogeneous composition over the cross-section of the ingot, five remeltings were performed with the overturning of the ingot. The mass loss during the melting process did not exceed 1%. Metallic alloy ribbons with a thickness of about 25 μm and a width of 1 mm were produced by the single rollermeltspinning process in an Ar atmosphere. The nitrogen content determined by the reduction melting method via LECO TC-600 Oxygen and Nitrogen Analizator in the investigated alloys was 0.04 ± 0.01 wt.%. The investigated alloys were divided into three groups, Mo-B, Mo,V-B, and V-B, which denote alloys $(Fe_{0.25}Ni_{0.25}Co_{0.25}Cr_{0.125}Mo_{0.125})_{100-y}B_y$, $(Fe_{0.25}Ni_{0.25}Co_{0.25}Cr_{0.125}Mo_{0.0625}V_{0,0625})_{100-y}B_y$, and $(Fe_{0.25}Ni_{0.25}Co_{0.25}Cr_{0.125}V_{0.125})_{100-y}B_y$, respectively. Individual alloys were designated as Mo-XXB, where XX was the boron content in the alloy.

An X-ray structural analysis was conducted using a Bruker D8 Advance diffractometer (Bruker, Germany). The diffraction patterns were obtained according to the Bragg-Brentano scheme using Cu Kα radiation with a monochromator on the reflected beam (Bruker, Germany). The microstructure was investigated using scanning and transmission electron microscopy, using Zeiss Merlin and Zeiss Libra 200 microscopes (Zeiss, Germany) equipped

with the EDX analyzers (Oxford Instruments, United Kingdom). Thermal properties were evaluated by a differential scanning calorimeter (DSC) (Setaram, France) method with a heating rate of 0.67°/s. The ribbons' microhardness was measured using a microhardness tester (Wilson&Wolpert, Netherlands) at a load of 980 mN.

## 3. Results and Discussion

### 3.1. Structure Investigation

Figure 1 shows the diffraction patterns of the investigated alloys. Alloy ribbons of the Mo-B group (Figure 1a) exhibited an amorphous structure. A wide diffuse maximum was observed in the range of 2θ angles from 35° to 52° on the diffraction pattern, regardless of the boron content in the alloy, and there were no sharp diffraction maximums associated with the formation of crystalline phases. Partial and complete replacements of molybdenum with vanadium led to the formation of a partially crystalline structure after quenching.

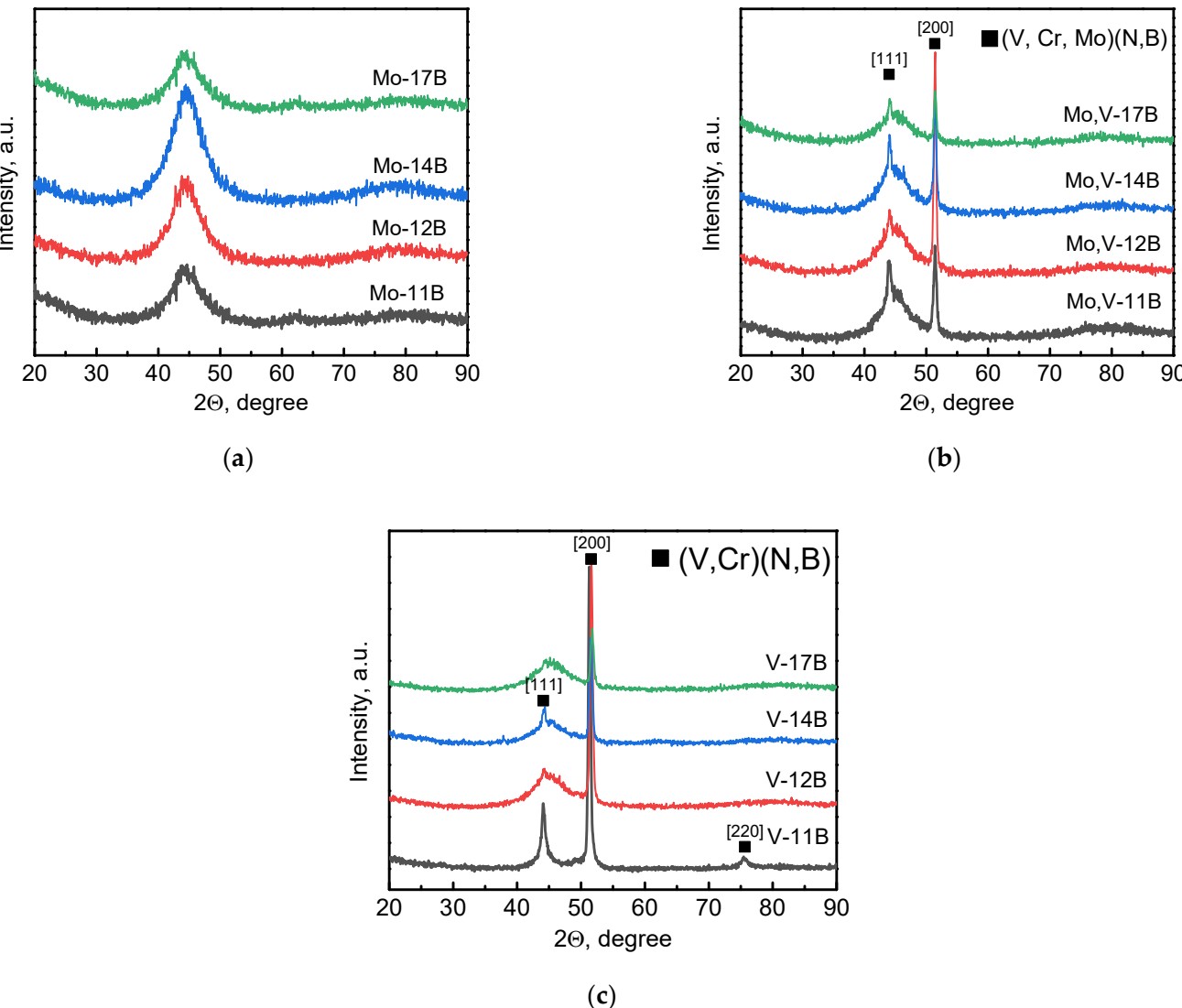

**Figure 1.** XRD patterns of alloy ribbons of (**a**) Mo-B, (**b**) Mo,V-B, and (**c**) V-B groups in cast state.

The diffraction patterns of the Mo, V-B, and V-B (Figure 1b,c) alloy ribbons showed a diffuse maximum corresponding to the amorphous phase and sharp X-ray reflections from the crystal planes corresponding to the formation of the fcc phase during quenching. There was a discrepancy between the relative intensities of the reflections of the families of planes, which was associated with the predominant orientation of the (200) planes parallel to the ribbon's surface. The (100) direction was the direction of the growth of the crystals with an fcc lattice, coinciding with the direction of heat removal, which, in this case, was directed perpendicular to the plane of the ribbon's surface [34]. An increase in the vanadium content in the alloys led to an increase in the content of the crystalline phase in the ribbon structure. This was evidenced by an increase in the relative intensity of X-ray reflections of the fcc phase towards the diffuse maximum.

An Increase in the boron content in the groups of Mo, V-B, and V-B alloys led to a decrease in the fraction of the crystalline phase. This was confirmed by a decrease in the relative intensity diffraction maximums of the fcc phase. It was impossible to obtain a single-phase amorphous structure for the Mo, V-B, and V-B alloy groups. The structures contain a crystalline fcc phase, regardless of the boron content. The comparable intensity of the diffraction maximums of the fcc phase and diffuse maximum indicated the small mass fraction of the crystalline phase. The planes of the fcc lattice had a high repeatability factor, directly proportional to their relatively high intensity compared to the diffuse maximum.

The structure of the cross-section of the ribbons was studied to determine the chemical composition of the crystalline phase formed during quenching. Figure 2 shows the SEM microstructure of the studied alloys and the X-ray microanalysis (EDX) spectrums.

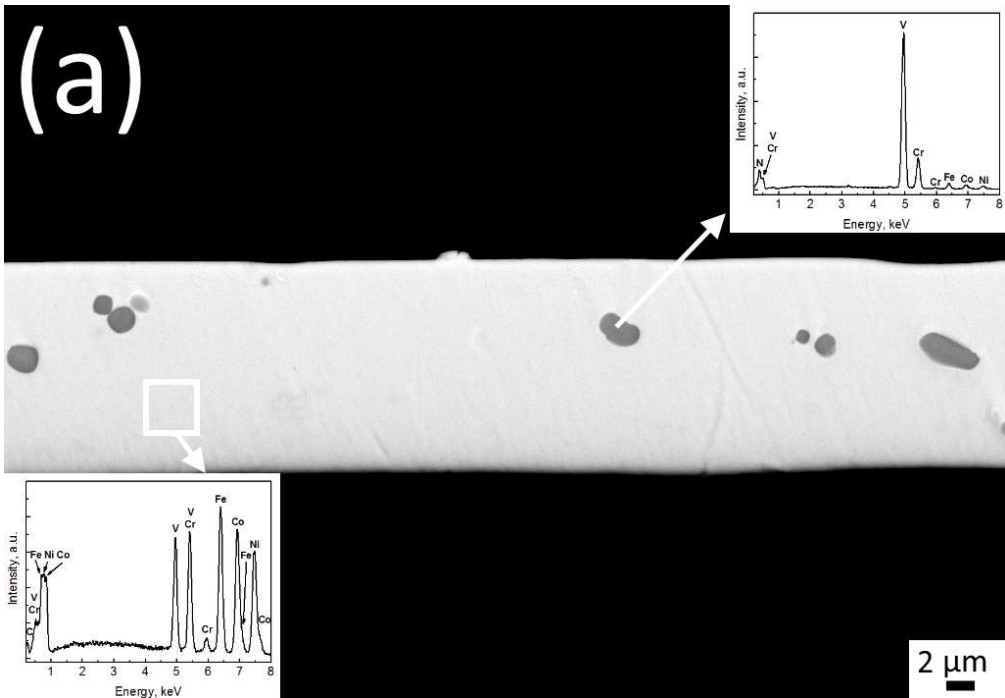

**Figure 2.** *Cont.*

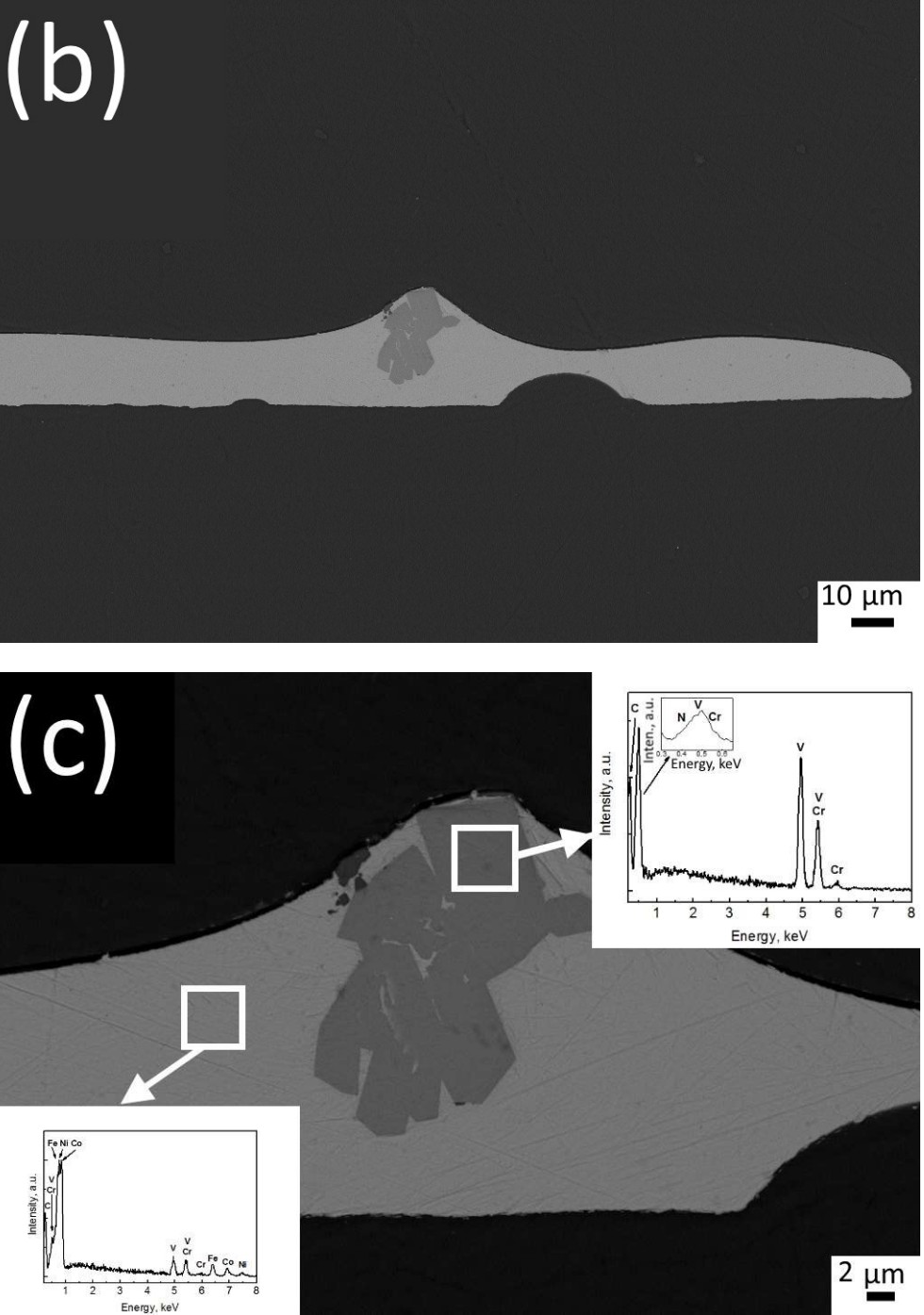

**Figure 2.** SEM images of the cross-section of the (**a**) V-11B and (**b**,**c**) V-17B alloy ribbons. Inserts are the EDX spectrums of the marked areas.

The SEM image of the V-11B alloy structure (Figure 2a) shows a homogeneous amorphous matrix phase of a light-gray color and inclusions of a crystalline phase with sizes from 1 to 5 μm. At a lower boron content, the crystalline phase presented a shape that is almost spherical and was located over the entire cross-section of the ribbon. The crystals of the V-17B alloy ribbon (Figure 2b) had a faceted structure and were located on the air surface. The crystal size varied from 10 to 19 μm. The EDX analysis of the crystalline regions (Figure 2c) showed that the formed crystals were rich in vanadium and chromium compared to the amorphous matrix. According to the EDX data, the crystalline phase in the alloy with a low boron content (V-11B) was rich in vanadium. In addition, the crystals

were characterized by an increased content of nitrogen, an element that was difficult to detect by the EDX method, which indicated its high content. In comparison, the XRD results indicate that they may belong to the VN (Fm3m)-type phase. An increase in the boron content led to the precipitation of crystals rich in vanadium and chromium, which indicated the formation of crystals in the (V,Cr)N phase with a similar lattice-type structure as the V-17B alloy.

Figure 3a shows an HRTEM image of the V-17B alloy ribbon exhibiting the boundary between a crystal and amorphous matrix. The presented selected area electron diffraction (SAED) patterns confirm the formation of a completely amorphous matrix structure (Figure 3b), while the diffraction pattern in the crystalline region (Figure 3c) indicates the formation of a crystal with an fcc lattice. The crystalline particle is rich in vanadium, chromium, nitrogen, and boron, according to the distribution map of the chemical elements (Figure 3d), while the amorphous matrix contains all the constituent elements. Thus, the TEM observation confirms the formation of amorphous matrix and fcc crystals.

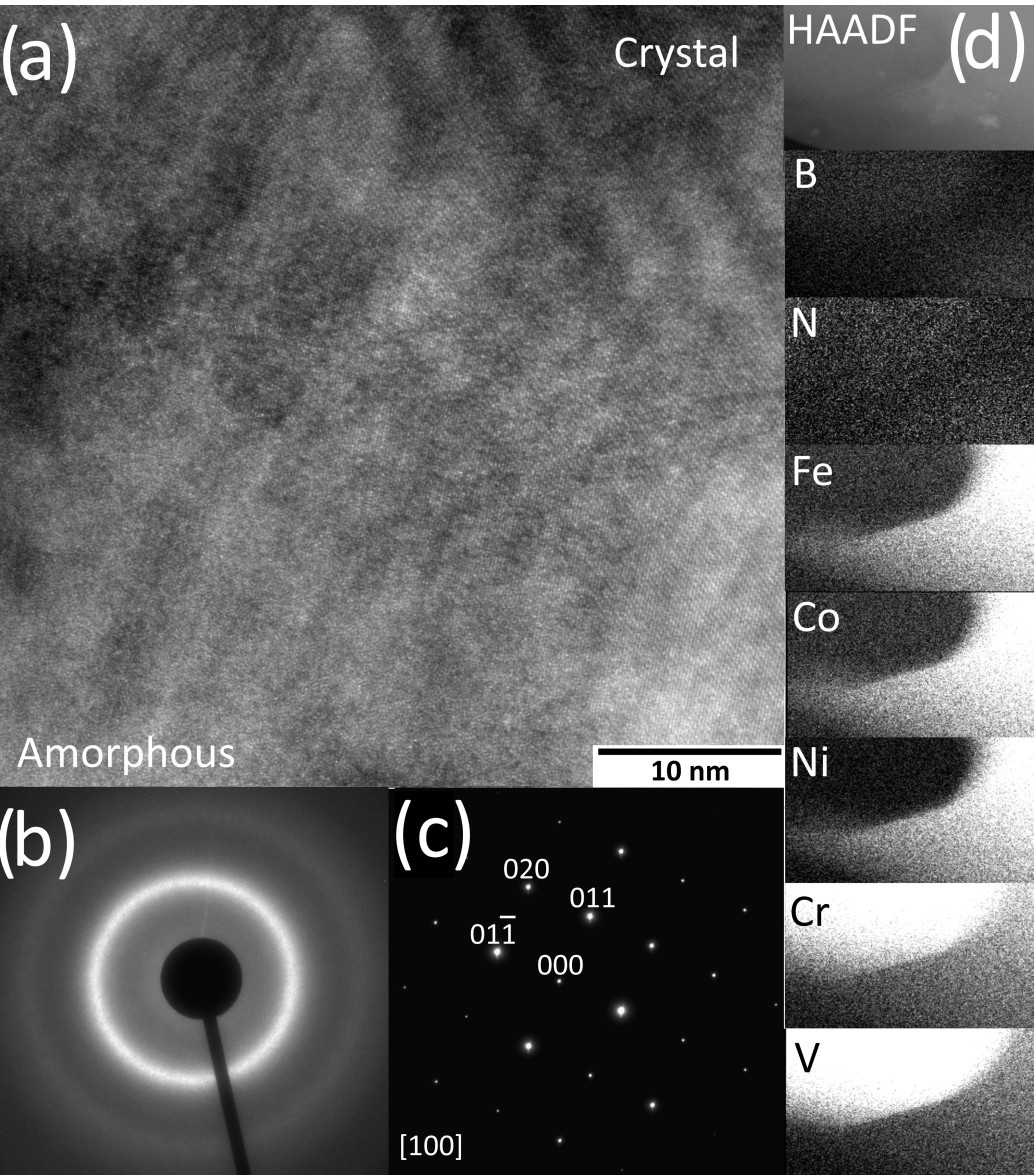

**Figure 3.** HRTEM image of V-17B alloy's structure (**a**), SAED from amorphous (**b**) and crystalline (**c**) areas, and high-angle annular dark-field (HAADF) image and EDX element's distribution map (**d**).

Unfortunately, it was difficult to reliably determine the stoichiometry of the formed crystals using EDX methods due to the inaccuracy of determining boron and nitrogen in

the composition. However, based on the obtained results and the analysis of the phase diagrams, it can be assumed that the observed crystals are vanadium and chromium nitrides with a VN and (V,Cr)N (Fm3m)-type structure. The formation of nitrides was indicated by the observed crystals, mainly containing V and Cr in their compositions; these metals are infinitely soluble in each other and possess a bcc lattice. Thus, these metals could not form a solid solution with an fcc lattice. In addition, these metals did not form intermetallic compounds with boron with a simple cubic lattice. At the same time, we observed the formation of an fcc phase in the ribbon structure and increased nitrogen and boron contents in these crystals. These metals formed stable compounds with nitrogen with either VN or CrN formulas. Nitrogen exists in dissolved form in raw metals, such as Cr, V, Fe, and Mo. Typically, the nitrogen content in alloys based on the Fe-Ni-Cr system varies at the level of 0.02–0.07 wt.%, and in special alloys, it can reach several tenths of a percent [35]. Hence, based on the data we obtained, we assumed that these crystals were nitrides.

### 3.2. Calorimetric Studies

Calorimetric studies of the alloys were conducted to establish the characteristic temperatures of the alloys and the effect of the addition of vanadium. Figure 4 shows the DSC curves of the studied alloys and the dependences of the characteristic temperatures on alloy composition.

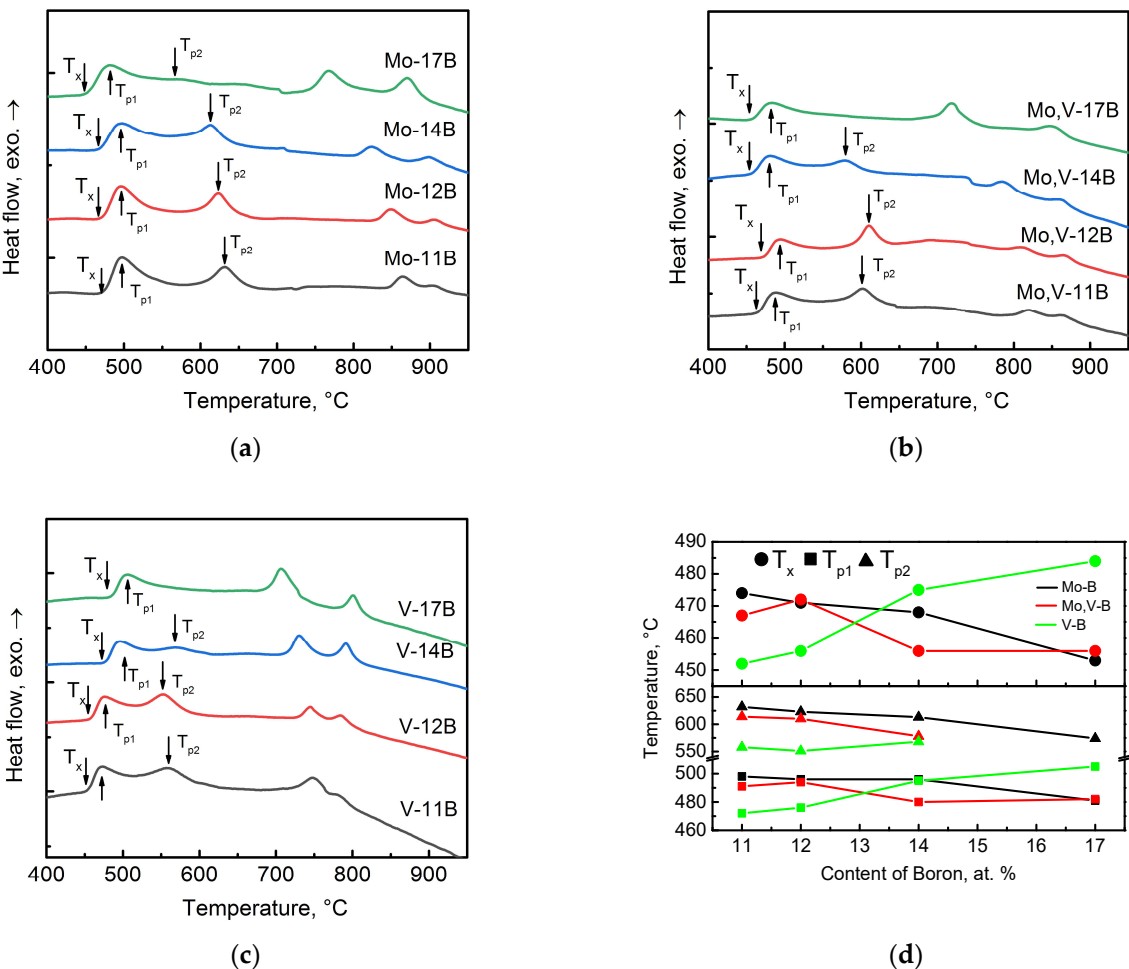

**Figure 4.** DSC curves of as-cast (**a**) Mo-B, (**b**) Mo,V-B, and (**c**) V-B alloys obtained with a heating rate of 0.67°/s. (**d**) Dependences of the characteristic temperatures on the boron content, where $T_x$ is the crystallization temperature, and $T_{p1}$ and $T_{p2}$ are the temperatures of the first and second peaks on the DSC curve, respectively.

The crystallization of the alloys started from the amorphous state, since the DSC curves (Figure 4a–c) lacked the endothermic effects associated with the glass transition process in the amorphous phase. All groups of alloys underwent similar crystallization processes determined by the boron content. In alloys with 11–14% B in the temperature range of 450–650 °C, crystallization presented a two-stage character with two exothermic peaks. An increase in the boron content up to 17% led to the merging of these processes into one long-term exothermic reaction (curves for Mo-17B; Mo,V-17B; V-17B in Figure 4a–c). For alloys in the Mo-B group, an increase in the boron content led to a decrease in the crystallization temperature ($T_x$) of the alloys (Figure 4a,d), which indicated the start of a structural change from metastable amorphous to crystalline states. The crystallization temperature of the Mo,V-B alloys changed with a maximum value at 12 at.%, presenting an additional sharp decrease up to 17 at.% B (Figure 4b,d). The alloys in the V-B group presented an increase in the crystallization temperature with an increase in the boron content (Figure 4c,d). It can be argued that for alloys containing molybdenum, an increase in the boron content leads to a decrease in the thermal stability of the initial amorphous matrix. At the same time, for alloys that completely replace molybdenum with vanadium, an increase in the boron content leads to an increase in $T_x$, i.e., an increase in the thermal stability of the amorphous phase (Figure 4c). The interval between $T_{p1}$ and $T_{p2}$ in Figure 4d disappears for Mo,V-17B and V-17B alloys.

### 3.3. Microhardness Analysis

Figure 5 shows the dependence of the microhardness of the alloys on the boron content. Partial and complete replacements of molybdenum by vanadium led to a change in microhardness dependencies. The Mo-B alloys demonstrated a decrease in microhardness with an increase in boron content, while vanadium-containing alloys exhibited the opposite dependencies—the hardness increased with an increase in the boron content. Additionally, vanadium containing low-boron alloys exhibited reduced hardness values. However, an increase in the boron content resulted in higher microhardness values compared to molybdenum-containing alloys.

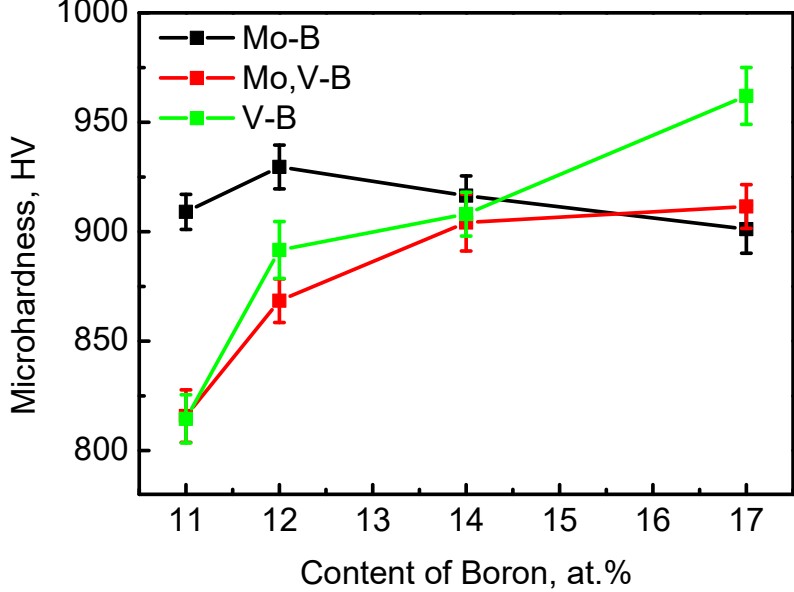

**Figure 5.** Dependencies of microhardness of the investigated alloys on boron content.

### 4. Discussion

Concerning the formation of nitride phases in the rapidly quenched quasi-high-entropy Fe-Co-Ni-Cr-(Mo,V)-B alloy ribbons, vanadium and chromium presented a very high affinity with nitrogen. Nitrogen was dissolved in the raw materials (in vanadium,

chromium, iron, and molybdenum) [36] and formed vanadium nitrides during melting, where the vanadium could be partially replaced by chromium. However, the alloys contain a high content of boron, which is a potent deoxidizer and nitride former [37]. Vanadium nitrides were observed in the ribbon structure; at the same time, the formation of boron nitrides was not observed. Since the Gibbs energy of boron nitride was much lower than that of vanadium nitride, the formation of boron nitrides during melting was thermodynamically more favorable. This was also confirmed by Ellingham diagrams via the plot of Gibbs free energy change ($\Delta G$) for the nitrides' formation by melting as a function of temperature [38]. The strongly negative mixing enthalpy of elements ($\Delta H_{mix}$) played a decisive role in the formation of vanadium nitrides, which were ($-143$) and ($-28$) kJ/mol for the V-N and B-N pair, respectively [39]. At the same time, it was essential to note that the formation of a crystalline phase in the structure of the ribbon was not associated with a lack of cooling during melt spinning. Vanadium nitrides are formed during the melting process and are thermally stable up to temperatures exceeding 2000 °C, while the casting temperature of the ribbons does not exceed 1500 °C. At a low cooling rate, fcc or bcc iron-based solid solutions are formed in similar alloys [31], depending on the content of $\gamma$-stabilizers in the composition of the solid solution. We assumed that the replacement of molybdenum by vanadium in the studied alloys caused a shift in the alloy compositions to the phase region of nitride formation, which was an unusual effect since the formation of borides was usually observed in similar alloys. It is known that the Fe-Ni-Cr ternary system is sensitive to the nitrogen content and formations in the structure of nitrides. The formation of chromium and vanadium nitrides is often observed in austenitic steels [40,41]. The investigated alloys had similar compositions containing high amounts of nickel and chromium.

In alloys with molybdenum, nitrogen is dissolved in a multicomponent solid solution, and the addition of vanadium leads to the formation of hardly soluble nitrides during melting, which can be observed in the ribbons' structure. Thus, vanadium acts as a getter that binds dissolved nitrogen from the melt to nitrides. Additionally, boron is highly soluble in zirconium and hafnium nitrides [42], which have a structure similar to vanadium and chromium nitrides. Thus, a large volume fraction of the observed nitrides in the structure occurred due to the replacement of nitrogen atoms by boron atoms. However, not all vanadium was consumed to form nitrides, and most of it was retained in the amorphous matrix, as evidenced by the EDX data. Based on the ratio of alloying elements in the composition of the amorphous phase, 7 to 13% of the vanadium added into the alloys was spent on the formation of vanadium nitrides. It was difficult to precisely determine the content of vanadium in an amorphous matrix, since it was impossible to determine the exact content levels of boron and nitrogen in the phases by EDX. The measuring error of the light-elements content could reach 50%. Therefore, these elements were eliminated from the spectrum analysis. Thus, the content of elements in the matrix could be estimated only from the ratio, comparing this with the similar ratio of the master alloy. The estimated mass fraction of nitrides did not exceed 5 wt.% considering the molar masses of the amorphous matrix and vanadium nitride.

In alloys containing molybdenum, an increase in the boron content led to a decrease in the thermal stability of the initial amorphous matrix. At the same time, in alloys with a complete replacement of molybdenum by vanadium, an increase in the boron content led to an increase in the crystallization temperature $T_x$, i.e., an increase in the thermal stability of the amorphous phase. All the studied groups of alloys showed a decrease in the temperature interval between the first and second stages of crystallization up to the transition to a single-stage type of crystallization. The intervals between $T_{p1}$ and $T_{p2}$ disappeared for Mo,V-17B and V-17B alloys. The addition of vanadium led to a decrease in this temperature range due to a decrease in the temperature at the beginning of the second crystallization stage. Thus, replacing molybdenum with vanadium decreased the thermal stability of the residual amorphous matrix after the first crystallization stage. Typically, the stability of the amorphous phase in iron-based alloys is highly dependent on the type of

crystallization. Thus, alloys exhibiting the primary crystallization of a solid solution have a lower crystallization temperature compared to alloys exhibiting the simultaneous growth of two or more phases [43].

The two-stage crystallization of amorphous alloys based on iron and amorphous high-entropy alloys based on Fe, Ni, Co with a metal content higher than 80 at% is associated with the low thermal stability of the amorphous phase. As compared with commercial iron-based amorphous alloys, the increased content of metals in the composition is the reason for the primary crystallization of the iron-based bcc solid solution [44,45]. The formation of primary metal-rich solid-solution crystals from the amorphous matrix led to the enrichment of the residual amorphous matrix with boron. Simultaneously, the residual amorphous matrix composition is shifted into a eutectic one.

The observed changes in the character of the dependence of the microhardness of the alloy can be associated with two factors—changes in the chemical composition of the amorphous phase and the presence of crystalline particles in the structure. A large proportion of the crystalline phase in alloys with vanadium and a low boron content led to the depletion of the amorphous matrix in boron, chromium, and vanadium, which were part of the observed crystals. Thus, an increase in the microhardness of alloys with vanadium with an increase in the boron content was associated with an increase in the volume fraction of the amorphous phase and an increase in the concentration of alloying elements in it. The deformation of metallic glasses was conducted by the nucleation and propagation of shear bands in the amorphous matrix [46]. The presence of crystalline phases of vanadium nitrides in the structures of Mo, V-B, and V-B alloys can affect the microhardness of alloys by blocking the shear bands' propagation and the nucleation of new ones. The deformation mechanism of crystalline particles differs, from the amorphous matrix. Thus, upon creating a crystalline particle, the shear band stops or multiplies; these processes lead to the multiplication of shear bands in the material and the transition to quasi-homogeneous deformation. This behavior of composite materials based on metallic glasses usually leads to a reduction in the strength of the material but significantly increases its plasticity [47–49].

## 5. Conclusions

The paper studied the effects of partial or complete replacements of molybdenum by vanadium in rapidly quenched quasi-high-entropy alloy ribbons with the compositions of $(Fe_{0.25}Ni_{0.25}Co_{0.25}Cr_{0.125}Mo_{0.125-x}V_x)_{100-y}B_y$, where x = 0; 0.0625; 0.125 and y = 11–17, containing an admixture of nitrogen at a concentration of $0.04 \pm 0.01$ wt.%, on the microstructure, phase composition, and microhardness.

1. It was established that replacing molybdenum with vanadium led to the formation of crystals located in the amorphous matrix and on the air surface of the ribbon, depending on the concentration of boron in the alloys. Based on the studies of the structure by the EDX, TEM, SEM, and XRD methods, it was assumed that the observed crystals were refractory nitrides based on vanadium V(N,B) and vanadium and chromium (V,Cr)(N,B) with a crystal structure (fm3m). The suggested reason for the formation of refractory nitrides was the strongly negative enthalpy of mixing vanadium with nitrogen. It was shown that an increase in the boron content led to a decrease in the volume fracture of vanadium nitride crystals in the structure.

2. It was shown that replacing molybdenum with vanadium led to an increase in the thermal stability of the initial amorphous matrix and a decrease in the temperature of the second crystallization stage. An increase in the boron content in alloys of the Mo-V group led to the decreased thermal stability of the amorphous matrix. In V-B alloys, an increase in the crystallization temperature was observed with an increase in the boron content.

3. The replacement of molybdenum with vanadium radically changed the nature of the alloy's microhardness dependencies on the boron content. The Mo-B alloys exhibited a decrease in hardness by 30 HV, while V-B alloys showed an increase in hardness from 815 to 960 HV with the increasing boron content.

**Author Contributions:** Conceptualization, A.B. and E.Z.; methodology, A.B.; validation, A.B. and E.U.; formal analysis, A.B.; investigation, I.S., E.U. and M.P.; resources, D.M.; data curation, A.B.; writing—original draft preparation, A.B.; writing—review and editing, E.Z.; visualization, I.S. and M.P.; supervision, A.B.; project administration, A.B. and D.M.; funding acquisition, A.B. All authors have read and agreed to the published version of the manuscript.

**Funding:** The ribbon sample preparation, microhardness testing, and XRD and DSC measurements were supported by the Russian Science Foundation, grant No. 22-79-10055. Master alloys' preparation was funded by the Strategic Academic Leadership Program "Priority 2030" (project K2-2022-001).

**Data Availability Statement:** The data presented in this study are available on request from the corresponding author.

**Acknowledgments:** The authors are grateful to the Interdisciplinary Resource Center "Nanotechnologies" of St. Petersburg State University for their assistance in conducting the SEM and TEM studies. For the TEM sample preparation, the authors thank the Collective Use Equipment Center "Material Science and Metallurgy" for the equipment modernization program represented by the Ministry of Higher Education and Science of Russian Federation (No.075-15-2021-696).

**Conflicts of Interest:** The authors declare no conflict of interest.

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
