# Peer review of "Structure and Properties of Amorphous Quasi-High-Entropy Fe-Co-Ni-Cr-(Mo,V)-B Alloys with Various Boron Content"

_metals, doi:10.3390/met13081464_

Round 1

Reviewer 1 Report

The authors investigate the vanadium substitution on quasei high entropy alloys. The manuscripts deals with the characterization of the alloys, thermal and mechanical properties measurements. Overall the manuscript is well written and the physical discussion of the results are clear. Therefore, the manuscript deserves publication. However, it still requires some improvements before it has my recommendation for publication on Metals.

1) The authors should better discuss some of the methods used. Sometimes, only a reference is not enough and some description should be given.

a) Please better describe the Erhenfest equation and the approximating the obtained diffuse maxima with Gaussian functions method.

b) Please describe the Ellingham diagrams.

2) What the authors mean by Crystallization temperature Tx? Is it the temperature the system change from amorphous to crystal? Is it reversible?

In conclusion, after the authors improve the manuscript considering my minor changes., It will be suitable for publication on Metals.

The overall quality of the english is good. I did not detect any typos or grammar mistakes.

Reviewer 2 Report

 please see the attached doc document 

Round 2

Reviewer 2 Report

Cosmetics improvemets concerning references:

ref 1,11,28 -use small typechar in author names

ref 6,24,26,27,31,33,35,37 - please regard use subscripts in compound chemical coding

ref 31,37 pease regard use of the "non-breakin space" in order to avoid a big distance , coming from layout process, in the compound label (name).

Author Response

Many thanks for pointing out the mistakes in list of references. All points have been corrected.